# Evaluation of Vitamin D Fractions in Obese Hypertensive Patients

**DOI:** 10.3390/ijerph17051660

**Published:** 2020-03-04

**Authors:** Marta Pelczyńska, Teresa Grzelak, Marcelina Sperling, Matylda Kręgielska-Narożna, Paweł Bogdański, Krystyna Czyżewska

**Affiliations:** 1Department of Treatment of Obesity, Metabolic Disorders and Clinical Dietetics, Poznan University of Medical Sciences, Szamarzewskiego 84 St., 60-569 Poznan, Poland; 2Department and Division of Physiology, Poznan University of Medical Sciences, Swiecickiego 6 St., 60-781 Poznan, Poland; 3Division of Biology of Civilization-Linked Diseases, Poznan University of Medical Sciences, Swiecickiego 6 St., 60-781 Poznan, Poland; 4Stanisław Staszic University of Applied Sciences in Pila, Podchorazych 10 St., 64-920 Pila, Poland

**Keywords:** vitamin D, free vitamin D, bioavailable vitamin D, obesity, hypertension

## Abstract

Vitamin D fractions can be involved in the pathogenesis of metabolic disorders, but their concentrations are rarely determined. The aim of this study was to evaluate the concentration of vitamin D fractions in obese hypertensive patients and to determine its associations with anthropometric parameters, glucose levels, and lipid profiles. A total of 85 obese hypertensive patients (OBHT) and 40 nonobese nonhypertensive subjects (NOBNHT) underwent biochemical measurements of lipid profiles, glycemia, 25-hydroxyvitamin D (25(OH)D), free vitamin D (free25(OH)D), vitamin D binding protein, albumin levels. Moreover, free25(OH)D and bioavailable25(OH)D (bio25(OH)D) concentrations were calculated. Blood pressure and anthropometric measurements were performed. Differences between groups (*p* < 0.001) were found for 25(OH)D (OBHT 40.25 ± 18.02 vs. NOBNHT 64.10 ± 22.29 nmol/L), free25(OH)D (9.77 (7.46; 11.49) vs. 13.80 (10.34; 16.82) pmol/L), bioavailable 25(OH)D (3.7 (2.8; 4.4) vs. 5.4 (4.2; 6.7) nmol/L), and calculated free25(OH)D (7.82 (5.54; 11.64) vs. 10.46(8.06;16.28) pmol/L, *p* = 0.002). The OBHT patients showed no relationship between vitamin D fractions concentration and glucose or lipids level, although it was associated with anthropometric parameters. In the NOBNHT group, vitamin D fractions correlated positively with HDL cholesterol and negatively with triglyceridemia and hip circumference. Vitamin D fractions were decreased in obese hypertensive subjects, and were associated with anthropometric parameters, but not with glucose level or lipid profiles; they thus cannot be considered as a predictive marker of metabolic disorders in this group of patients.

## 1. Introduction

Excessive body weight is perceived as an independent risk factor for the development of hypertension, and the population of obese hypertensive people continues to increase. In recent years, particular attention has been paid to the role of vitamin D fractions in the pathogenesis and the course of obesity. It is believed that a decreased serum concentration of vitamin D fractions is associated with a higher risk of metabolic disorders, such as insulin resistance and dyslipidemia, in this group of patients. One factor that affects vitamin D concentration is the amount of visceral adipose tissue, which sequesters this vitamin, affecting its concentration in the blood [1,2].

The principal method used to evaluate vitamin D concentration is to determine the total level of 25-hydroxyvitamin D (25(OH)D), that is, 25-hydroxyergocalciferol and 25-hydroxycholecalciferol, since the 1.25-dihydroxyvitamin D form is labile [3,4]. However, in the human body, there are several forms of vitamin D. Most fractions of this vitamin (85%–90%) are strongly bound to vitamin D binding protein (VDBP), which is the protein responsible for transporting it. About 10%–15% of vitamin D is bound to albumin, and it is estimated that there are only very small amounts (about 0.03%) of the vitamin in the free fraction of the total pool. This free fraction, together with the vitamin D bound with albumin, forms the bioavailable fraction [5]. The free hormone hypothesis assumes that only molecules not bound to serum proteins can migrate through cell membranes (because of its high biological activity), which would mean that determination of the free (free25(OH)D) and bioavailable (bio25(OH)D) fractions of vitamin D should be performed alongside current methods of assessing vitamin D status [6].

Despite widespread hypovitaminosis D, the status of this vitamin in the human body is relatively rarely determined in clinical practice. One parameter assessed in experimental studies is the concentration of 25-hydroxyvitamin D. In some studies, the concentration of free vitamin D has been described, but based solely on mathematical calculations and not on biochemical analysis. To date, there have been no comprehensive studies assessing the concentration of vitamin D fractions, especially in obese hypertensive patients. On one hand, some analyses have indicated an association between decreased concentrations of vitamin D fractions and the occurrence of metabolic disorders, such as diabetes or dyslipidemia [7,8]. Moreover, some other studies indicate an inverse relationship between vitamin D status and hypertension in obese subjects [9,10]. On the other hand, the cause-and-effect relationship between these has not been explicated, and for this reason we undertook this study.

The aim of our research was to evaluate the concentration of vitamin D fractions in obese hypertensive subjects and to determine its associations with anthropometric parameters, glucose level, and lipid profile in this group of patients.

## 2. Materials and Methods

### 2.1. Research Group

The research was approved by the local bioethics commission (approvals 456/14 and 729/17). Participation in the study was voluntary, and each person signed informed consent. The study was conducted in accordance with the Helsinki Declaration.

A total of 125 Caucasian individuals aged 30–60 years were examined. Of 105 (Figure 1) obese hypertensive patients, 85 people (36 women and 49 men) aged 46.9 ± 8.5 years were included in the study as the research group (OBHT). The inclusion criteria for the OBHT group were body mass index (BMI) ≥ 30 kg/m^2^ (obese) or waist circumference above 80 cm for women and 94 cm for men, stable body weight for one month before the trial (±1 kg), hypertension (blood pressure value above 140/90 mmHg [11]), and Caucasian. The obese hypertensive people were all patients at the Department of Internal Medicine, Metabolic Disorders and Hypertension, Poznan University of Medical Sciences. In addition, 40 nonobese volunteers (21 women and 19 men with BMI between 18.5 and 24.9 kg/m^2^ or waist circumference under 80 cm for women and 94 cm for men), without hypertension, aged 42.4 ± 7.4 years were included in the analysis as the NOBNHT group. Recruitment for both groups was conducted simultaneously for 29 months, beginning in February 2013. Participants were included in the study in a consecutive manner.

The exclusion criteria were secondary obesity or secondary hypertension; hepatic, renal, adrenal, or thyroid disorders; a positive history of oncological or autoimmune diseases; alcoholism or acute symptoms of infection in the three months prior to the study; insulin therapy; and taking any dietary supplements in the six months prior to the analysis. In addition, none of the individuals had any limitations that could affect endogenous synthesis of vitamin D (such as immobilization, frequent hospitalization, or current or previous steroid use). The study did not include people with sex hormone treatment, pharmacological treatment of hypertension, and pregnant or lactating women because of the risk of unreliable results.

### 2.2. Anthropometric and Blood Pressure Measurements

Patients were overnight fasting and dressed only in light clothes during anthropometric measurements. Body mass was measured by a certified electronic scale (SECA 285 Wireless, Hamburg, Germany) with an accuracy of 0.1 kg, and height was measured using a stadiometer with an accuracy of 0.1 cm. Waist circumference was measured at the midway between the costal arch and the upper iliac crest and hip circumference at the level of the greater trochanters. These measurements were used to calculate the Body Mass Index. BMI was calculated as weight divided by height squared (kg/m^2^). In addition, body composition measurements were performed: Body adipose tissue, lean body mass, and water content were measured using the electrical bioimpedance method with a Bodystat 1500 (Bodystat Ltd., Douglas, UK). After a ten-minute sitting rest in accordance with the guidelines of the European Society of Hypertension and the European Society of Cardiology [12], blood pressure was measured three times, in the morning with the patient sitting with a certified automated blood pressure monitor (Omron, Kyoto, Japan).

### 2.3. Biochemical Measurements and Calculations

Blood samples of about 5 mL were taken in the morning (7:00–8:00) after overnight fasting and all-night rest, and after lying in a supine position for 30 minutes. Most of the biochemical analyses were performed immediately after collection, while the serum samples needed to measure free25(OH)D and VDBP were separated, secured, and frozen at −20 °C. The total concentration of 25(OH)D (25-hydroxyergocalciferol and 25-hydroxycholecalciferol) was evaluated using electrochemiluminescence on a Cobas e immunoanalyzer (Roche Diagnostic, Mannheim, Germany) with a limit of detection (LOD) 7.5 nmol/L and coefficient of variations (CV) < 6.2%. Serum free hydroxyvitamin D (CV < 10%) and vitamin D binding protein (CV < 5%) concentrations were determined by ELISA (Future Diagnostics Solutions, Wijchen, Netherlands; DRG Instruments, Marburg, Germany, respectively) in line with the manufacturer’s guidelines. The concentration of free25(OH)D in serum was measured in pg/mL and converted to pmol/L by using the formula: 1 pg/mL = 2.5 pmol/L, and the serum concentration of VDBP was measured in mg/dL and similarly converted to µmol/L by using formula 1 mg/dL = 0.194 µmol/L. An automatic technique using bromocresol green was used to measure albumin concentrations (Abbott Laboratories, Irving, TX, USA). Fasting blood glucose (FBG) and lipid profile—that is, total cholesterol (TC), high density lipoprotein (HDL), and triglyceride (TG)—were determined using enzymatic methods with standardized commercial tests (Cobas c, Roche Diagnostic, Mannheim, Germany). Low density lipoprotein (LDL) concentration (mmol/L) was calculated as LDL = TC − (HDL + TG/2.2), because triglyceridemia was lower than 4.52 mmol/L [13]. The concentration of the free fraction of 25(OH)D was calculated following the formula free25(OH)D [pmol/L] = 25(OH)D [nmol/L] / 1 + (6 × 10^3^ × [albumin µmol/L]) + (7 × 10^8^ × [VDBP nmol/L]) [14]. Based on the concentration of the free hydroxyvitamin D directly determined in serum, the concentration of bioavailable hydroxyvitamin D was calculated using the equation: bioavailable 25(OH)D [nmol/l] = (K_alb_ × [albumin µmol/L] + 1) × [free25(OH)D pmol/L], where K_alb_ is a constant for 25(OH)D binding with albumin (6 × 10^5^ [mol^−1^]) [15]. The anthropometric and biochemical characteristics of the OBHT and NOBNHT groups are shown in Table 1.

### 2.4. Statistical Analysis

The data are presented as means ± SDs, or as medians with upper and lower quartiles. Statistical analysis of the results was carried out using Statistica 12.5 software with Medical Set (StatSoft, Tulsa, OK, USA), including elements of descriptive statistics and statistical procedures, such as correlation analysis (Pearson’s for parametric distributions and Spearman’s for nonparametric distributions), as well as analysis of variance. Comparisons between groups were performed using the Mann–Whitney U-test, or the unpaired *t*-test if the data were normally distributed. In addition, the Wilcoxon test was used for groups of dependent variables. For the groups with less than 50 members, normal data distribution was determined using the Shapiro–Wilk test; we used the Kolmogorov–Smirnov test for larger groups. We determined that a sample size of at least 40 subjects in each group would yield at least an 80% power of detecting an effect as statistically significant at the 0.05 α level.

## 3. Results

The mean concentration (±SD) of 25(OH)D in all participants was 47.9 ± 22.4 nmol/L; it was significantly lower in the OBHT group, at 40.6 ± 18.0 nmol/L, than in the NOBNHT subjects (64.1 ± 22.3 nmol/L; *p* < 0.001; Figure 2). According to the 2011 recommendations of the Endocrine Society [16], this level represents vitamin D deficiency in obese hypertensive patients (vitamin D concentration in the range 25–50 nmol/L).

A similar relationship was seen for the level of free25(OH)D determined directly in the serum (9.8 (7.5; 11.5) vs. 13.8 (10.3; 16.8) pmol/L; *p* < 0.001; Figure 3), as well as for its bioavailable fraction (3.7 (2.8; 4.4) vs. 5.4 (4.2; 6.7) nmol/L; *p* < 0.001), which were lower in OBHT patients than in the NOBNHT group. Significant statistical differences between the groups also occurred in the calculated free25(OH)D (7.8 (5.5; 11.6) vs. 10.5 (8.1; 16.3) pmol/L; *p* = 0.002; Table 2). In addition, nonobese nonhypertensive individuals had a median difference (*p* = 0.029) for related data, namely; the free25(OH)D fractions (directly measured in serum and calculated). The directly measured value of free vitamin D was higher (about 3.3 pmol/L) than the calculated value. There was no such association in the OBHT group.

In the entire study population, the concentration of 25(OH)D was significantly correlated with anthropometric and biochemical parameters (negatively with body weight: r = −0.42, *p* = 0.001; waist circumference: r = −0.47, *p* = 0.001; BMI: r = −0.46, *p* = 0.001; TG: r = −0.34, *p* = 0.001; and FBG: r = −0.26, *p* = 0.004; and positively with HDL cholesterol: r = 0.44, *p* = 0.001). Similar correlations were observed for both the directly measured free25(OH)D (correlating negatively with body weight: r = −0.29, *p* = 0.001; waist circumference: r = −0.33, *p* = 0.001; BMI: r = −0.36, *p* = 0.001; TG: r = −0.26, *p* = 0.004; and FBG: r = −0.26, *p* = 0.004; and positively with HDL cholesterol: r = 0.29, *p* = 0.001) and the bioavailable fraction of vitamin D (correlating negatively with body weight: r = −0.30, *p* = 0.001; waist circumference: r = −0.34, *p* = 0.001; BMI: r = −0.37, *p* = 0.001; TG: r = −0.27, *p* = 0.002; and FBG: r = −0.25, *p* = 0.002; and positively with HDL cholesterol: r = 0.28, *p* = 0.001). The calculated concentration of free25(OH)D was correlated with a smaller number of anthropometric and biochemical parameters (negatively with body weight: r = −0.19, *p* = 0.036; waist circumference: r = −0.26, *p* = 0.004; BMI: r = −0.26, *p* = 0.004; and triglyceridemia: r = −0.35, *p* = 0.001; and positively with HDL cholesterol: r = 0.34, *p* = 0.001). Table 3 shows the correlation analysis.

The obese hypertensive patients showed no relationship between the concentrations of vitamin D fractions and biochemical parameters. Of the anthropometric parameters the concentration of 25(OH)D negatively correlated with hip circumference (r = −0.22, *p* = 0.047) and percentage of adipose tissue (r = −0.22, *p* = 0.045), and positively with percentage of lean body mass (r = −0.22, *p* = 0.042). Similar dependences were observed for the measured concentration of free vitamin D and its bioavailable form, which negatively correlated with percentage of adipose tissue (r = −0.29, *p* = 0.008 for free25(OH)D and r = −0.30, *p* = 0.005 for bio25(OH)D) and positively with percentage content of lean body mass (r = −0.28, *p* = 0.008 for free25(OH)D and r = −0.30, *p* = 0.005 for bio25(OH).

In the NOBNHT group in turn, the concentration of 25(OH)D and the calculated free25(OH)D fraction correlated positively with HDL cholesterol (r = 0.45, *p* = 0.001 and r = 0.48, *p* = 0.001, respectively for 25(OH)D and the calculated free25(OH)D) and negatively with triglyceridemia (r = −0.36, *p* = 0.001 and r = −0.54, *p* = 0.001, respectively). In addition, the concentration of the directly measured free25(OH)D was again positively correlated with HDL cholesterol (r = 0.38, *p* = 0.001). Moreover, the concentration of 25-hydroxyvitamin D (r = −0.39, *p* = 0.014) and of free calculated vitamin D (r = −0.33; *p* = 0.039) correlated negatively with hip circumference.

## 4. Discussion

Recent years have highlighted a relationship between vitamin D deficiency in the human body and the occurrence of obesity and hypertension. The current approach to this relationship involves determining 25(OH)D levels but does not include a complex assessment involving measurement (or calculation) of free25(OH)D, or calculation of the bioavailable fraction of the vitamin. From the clinical point of view, such measurements would give more complete information on vitamin D status [17], which we have demonstrated in our previous study of patients with metabolic syndrome (including only calculated vitamin D fractions) [18].

The present study documents lower concentrations of 25(OH)D, directly measured free25(OH)D and its bioavailable fraction, and calculated free25(OH)D in obese hypertensive patients than in nonobese nonhypertensive individuals. A decrease in total 25(OH)D has been documented in obese people with metabolic disorders in many clinical studies [7,19,20,21]. An important factor in this relationship is the interaction between vitamin D concentrations and the amount of adipose tissue. On one hand, adipocytes have the ability to interact with the enzymes involved in the metabolism of vitamin D (25-hydroxylase and 1α-hydroxylase) by lowering their activity [22]. On the other hand, it is believed that vitamin D may regulate the process of adipogenesis by promoting adipocyte apoptosis (in the case of murine cell line 3T3-L1) or adipogenesis activation to produce new insulin-sensitive adipose tissue (in the case of humans) [23]. In addition, the excess of adipocytes in obese hypertensive individuals may act as a storage facility that hinders the entry into the bloodstream of vitamin D synthesized in the cell membranes of the keratinocytes and dermal fibroblasts (the sequestration problem) [24].

The nonobese nonhypertensive group showed median differences for related data, such as the concentration of directly measured free25(OH)D and of calculated free25(OH)D; the laboratory value was 24% higher than the calculated value. Discrepancies between these variables were also found by Sollid et al. [25], and by Oleröd et al. [26]. To date, the free25(OH)D fraction has usually been calculated from the equation of Bikle et al. [14], based on the recalculation of 25(OH)D, albumin, and VDBP concentrations. It is worth noting that this formula was based on an equation developed for calculating the concentration of free testosterone, which is many times higher than that of free25(OH)D. In addition, laboratory-based methods of evaluating the level of free25(OH)D, which mainly use ultrafiltration, are labor-intensive and time-consuming, and are rarely used in clinical practice. More recently, serum free25(OH)D concentrations have been determined accurately by a method based on immunoenzymatic reactions [6,25,27]. The calculated concentrations of the free25(OH)D fraction, despite their clinical utility, are approximate in nature and may differ from the directly measured free25(OH)D. This was the case for the NOBNHT group, though not for the OBHT group. It can be assumed that the lack of difference in the directly measured and estimated concentrations of free25(OH)D in the obese hypertensive individuals may be related to the amount of adipose tissue that sequesters vitamin D, as well as affecting its volume dilution [28,29,30].

Decreased levels of vitamin D are associated with a higher risk of hypertension. In prospective studies evaluating four large cohorts, as well as in the NHANES-III analysis, a negative relationship between the vitamin D concentration and blood pressure values was observed [10,31]. It is believed that vitamin D may have specific protective properties. Studies in both humans and animals have shown that vitamin D inhibits the activity of the renin–angiotensin–aldosterone system by suppressing the renin gene, and more precisely by affecting the cis-DNA elements within the promoter of that gene. It thus exerts an antihypertensive effect [32,33].

Antisclerotic activity of vitamin D involves inhibition of the formation of foam cells, which it achieves by limiting cholesterol intake by macrophages; it also increases the transport of HDL cholesterol [34,35]. In addition, vitamin D has the ability to modify the processes of lipogenesis and lipolysis, as well as the activity of the PPAR-δ (peroxisome proliferator-activated receptor-δ) transcription factor, which is involved in the regulation of lipid metabolism [36]. In turn, the contribution of vitamin D to the pathogenesis of type-2 diabetes is not fully understood, although the localization of vitamin D receptors on pancreatic beta cells demonstrates the potential role for insulin secretion, thereby maintaining glucose levels [35,37].

In our entire study population, the hydroxylated form of vitamin D, its directly measured free fraction, and the bioavailable form correlated negatively with metabolic parameters such as abnormal triglyceride and glucose levels, while correlating positively with HDL cholesterol level. On the other hand, the calculated free25(OH)D correlated only with blood lipid parameters (positively with HDL cholesterol level and negatively with TG concentration). It is worth noting that concentrations of 25(OH)D and free25(OH)D in the NOBNHT group, whether directly measured or calculated, correlated positively with HDL cholesterol concentration and, in the case of the free calculated fraction alone, negatively with triglycerides. Other studies have indicated a negative relationship between the total concentration of 25-hydroxyvitamin D and the occurrence of metabolic disorders, such as excessive body weight, increased amounts of adipose tissue, dyslipidemia, hypertension, and elevated glycemia in both normal-weight and obese people [20,38,39]. In our OBHT group, such associations were found only for the anthropometric parameters, and not for the biochemical ones. On the other hand, vitamin D fractions may be involved in the complex pathogenesis of lipid metabolism in nonobese nonhypertensive subjects.

Currently, limited data are available on the correlation between free25(OH)D or bio25(OH)D fractions and glycemia or lipid profile in obese hypertensive people. Moreover, only a few studies have evaluated the usefulness of determining various fractions of this vitamin in assessing its status in the human body. In one study of obese and overweight people, in comparison with normal weight subjects, body mass index was negatively correlated with total 25(OH)D, as well as the free fraction of vitamin D [21]. In another study involving 1189 nondiabetic subjects, free and total 25(OH)D levels were positively associated with acute insulin response and glucose disposition index; however, after adjustment for BMI, only the free fraction of vitamin D was found to be related to the secretion of insulin [40]. In turn, Jorde reviewed the relationship between 25(OH)D, free25(OH)D, and VDBP concentrations and the occurrence of type-2 diabetes, finding that measures of total 25(OH)D as well as of the free fraction of this vitamin in diabetic patients may serve as new biomarkers for the development and treatment of this disorder [41]. In our study, the obese hypertensive patients did not show any correlations between vitamin D fractions and glucose levels, so we cannot confirm these findings.

Anthropometric analysis may also be useful in assessing the presence of vitamin D deficiencies in the human body. In our study, we demonstrated negative correlations between the concentration of vitamin D fractions and BMI, waist circumference, and percentage of adipose tissue, and a positive correlation with percentage of lean body mass (depending on the group considered). These results are consistent with those of other researchers. In one study, the strong predictive value of BMI was shown in relation to the determination of vitamin D status [42]. In another, deficiency of this vitamin was associated with higher body mass index, waist to hip ratio, and amount of adipose tissue [43].

This study has some limitations. First, blood samples were collected from people with varying degrees of obesity. The study group contained only Caucasians, so the results of this study should not be generalized to the general population. Another limitation is represented by the small size of the study and control groups, which means there were difficulties in dividing them into smaller subgroups that might take into account the direct impact of solar radiation on the production of vitamin D. The gold standard for assessing vitamin D status is the liquid chromatography-mass spectrometry. Due to the high costs of this method, electrochemiluminescence technique was used in this study. Nevertheless, this method is widely used in clinical trials.

## 5. Conclusions

Biochemical differences in obese hypertensive subjects include blood concentrations of 25-hydroxyvitamin D and its free fraction (both measured and calculated), as well as the bioavailable fraction, all of which were significantly lower in this group. Vitamin D fractions in such patients were associated only with anthropometric parameters, and there was no correlation with glucose level or lipid profile; they thus cannot be considered as direct predictive markers of metabolic disorders in such individuals. In contrast, in nonobese nonhypertensive people, they did correlate with blood lipid parameters. They may thus be important in the complex pathogenesis of lipid metabolism in these subjects. Nevertheless, it is worth considering the assessment of vitamin D status in the body, not only by determining the 25-hydroxyvitamin D concentration but also by laboratory measurement of the free25(OH)D fraction, which has high biological activity. Due to the prevalence of hypovitaminosis D, it is worth to consider a year-round supplementation of this vitamin, not only in people over 60 years, but also in the younger population, and especially in obese hypertensive subjects, which could contribute to improving their metabolic profiles.

## Figures and Tables

**Figure 1 ijerph-17-01660-f001:**
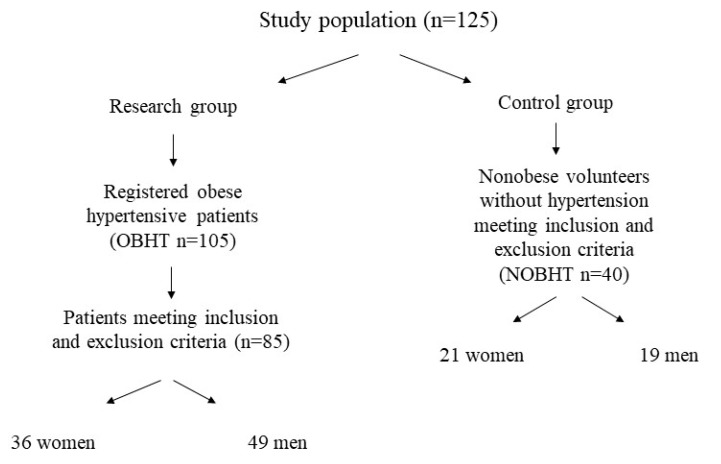
Patient recruitment to the study.

**Figure 2 ijerph-17-01660-f002:**
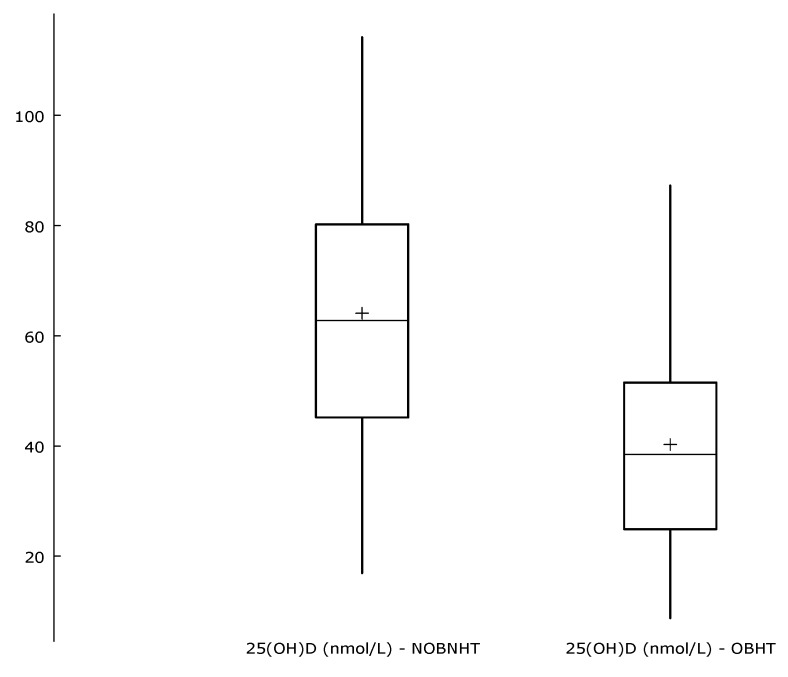
Serum concentration of 25-hydroyvitamin D in obese hypertensive individuals (OBHT) and nonobese nonhypertensive subjects (NOBNHT). These plots display the distribution of a variable. The central box encloses the middle 50% of the data, i.e., it is bounded by the first and third quartiles. The “whiskers” extend from each end of the box for a range equal to 1.5 times the interquartile range. A “+” sign is used to indicate the mean value.

**Figure 3 ijerph-17-01660-f003:**
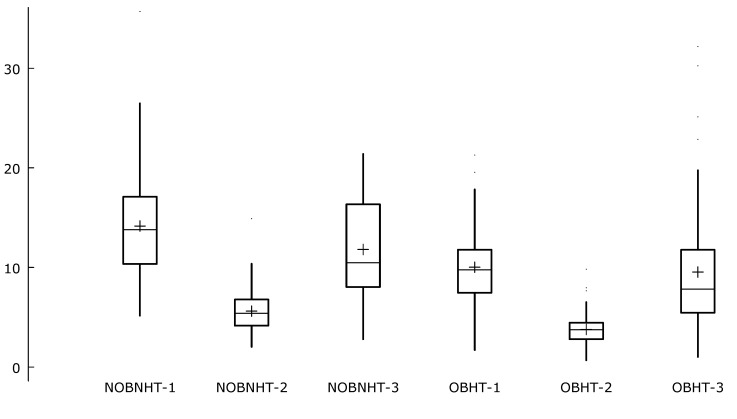
Concentration of vitamin D fractions in obese hypertensive individuals (OBHT) and nonobese nonhypertensive subjects (NOBNHT). NOBNHT-1—concentration of free25(OH)D in NOBNHT group (pmol/L); NOBNHT-2—concentration of bio25(OH)D in NOBNHT group (nmol/L); NOBNHT-3—concentration of calculated free25(OH)D in NOBNHT group; OBHT-1—concentration of free25(OH)D in OBHT group (pmol/L); OBHT-2—concentration of bio25(OH)D in OBHT group (nmol/L); OBHT-3—concentration of calculated free25(OH)D in OBHT group. These plots display the distribution of a variable. The central box encloses the middle 50% of the data, i.e., it is bounded by the first and third quartiles. The “whiskers” extend from each end of the box for a range equal to 1.5 times the interquartile range. A “+” sign is used to indicate the mean value.

**Table 1 ijerph-17-01660-t001:** Anthropometric and biochemical characteristics of obese hypertensive individuals (OBHT) and nonobese nonhypertensive subjects (NOBNHT).

Parameter (Unit)	OBHT (*n* = 85)	NOBNHT (*n* = 40)	*p* Value
Age (year)	46.9 ± 8.5	42.4 ± 7.4	0.089
Body weight (kg)	112.3 ± 22.2	72.5 ± 14.6	<0.001
Height (cm)	173 ± 9.0	171 ± 10.0	0.239
Waist circumference (cm)	117.9 ± 14.4	83.4 ± 10.8	<0.001
Hip circumference (cm)	119.1 ± 12.9	99.6 ± 6.6	<0.001
BMI (kg/m^2^)	37.4 ± 6.2	24.6 ± 3.1	<0.001
Body fat mass (%)	42.8 ± 8.4	24.7 ± 6.1	<0.001
Body lean mass (%)	57.2 ± 8.4	75.3 ± 6.1	<0.001
Water content in the body (%)	40.7 ± 5.0	54.8 ± 5.3	<0.001
SBP (mmHg)	150 (140; 160)	124 (106; 132.5)	<0.001 *
DBP (mmHg)	95 (87; 105)	80 (74.5; 90.25)	<0.001 *
TC (mmol/L)	5.5 ± 1.3	5.3 ± 0.8	0.275
HDL (mmol/L)	1.1 (0.9; 1.3)	1.8 (1.3; 1.9)	<0.001 *
LDL (mmol/L)	3.6 ± 1.2	3.3 ± 0.8	0.192
TG (mmol/L)	2.1 (1.6; 2.6)	1.0 (0.8; 1.4)	<0.001 *
FBG (mmol/L)	5.7 (5.2; 6.1)	5.0 (4.6; 5.4)	<0.001 *
Albumin (μmol/L)	624.9 ± 35.1	660.8 ± 38.3	<0.001
VDBP (μmol/L)	6.6 (5.1; 9.1)	7.8 (6.7; 9.2)	0.042

The values of parameters are shown as mean (±SD) or as medians (25% and 75% quartile); n—number of studies individuals; *p*—statistical significance level for OBHT vs. NOBNHT groups according to *t*-test for parametric data (possibly Welch test for differences in variance) or Mann–Whitney U-test * for nonparametric distributions; BMI—Body Mass Index; SBP—systolic blood pressure; DBP—diastolic blood pressure; TC—concentration of total cholesterol; HDL—concentration of high density lipoprotein; LDL—concentration of low density lipoprotein; TG—concentration of triglycerides; FBG—concentration of blood glucose; VDBP—concentration of vitamin D binding protein.

**Table 2 ijerph-17-01660-t002:** Concentration of various fractions of vitamin D in obese hypertensive individuals (OBHT) and nonobese nonhypertensive subjects (NOBNHT).

Parameter (Unit)	OBHT (*n* = 85)	NOBNHT (*n* = 40)	*p* Value
25(OH)D (nmol/L)	40.6 ± 18.0	64.1 ± 22.3	<0.001
Directly measured free25(OH)D (pmol/L)	9.8 (7.5;11.5)	13.8 (10.3;16.8)	<0.001
Bio25(OH)D (nmol/L)	3.7 (2.8;4.4)	5.4 (4.2;6.7)	<0.001
Calculated free25(OH)D (pmol/L)	7.8 (5.5;11.6)	10.5 (8.1;16.3)	0.002

The values of parameters are shown as mean (±SD) or as medians (25% and 75% quartile); n—number of studies individuals; *p*—statistical significance level for OBHT vs. NOBNHT groups according to *t*-test for parametric data (possibly Welch test for differences in variance); 25(OH)D—concentration of 25-hydroxyvitamin D; Bio25(OH)D—concentration of bioavailable 25(OH)D calculated based on directly determined free25(OH)D.

**Table 3 ijerph-17-01660-t003:** Correlation analysis (Pearson’s for parametric distributions or Spearman’s for nonparametric distributions) between variables across the entire studied population (*n* = 125).

Parameter (Unit)	Body Weight (kg)		Waist Circumference (cm)		BMI (kg/m^2^)		HDL (mmol/L)		TG (mmol/L)		FBG (mmol/L)	
	r_p_/r_s_ *	*p*	r_p_/r_s_ *	*p*	r_p_/r_s_ *	*p*	r_p_/r_s_ *	*p*	r_p_/r_s_ *	*p*	r_p_/r_s_ *	*p*
25(OH)D (nmol/L)	−0.42	0.001	−0.47	0.001	−0.46	0.001	0.44 *	0.001	−0.34	0.001	−0.26	0.004
Directly measured free25(OH)D (pmol/L)	−0.29	0.001	−0.33	0.001	−0.36	0.001	0.29 *	0.001	−0.26	0.004	−0.26	0.004
Bio25(OH)D (nmol/L)	−0.30	0.001	−0.34	0.001	−0.37	0.001	0.28 *	0.001	−0.27	0.002	−0.25	0.004
Calculated free25(OH)D (pmol/L)	−0.19 *	0.036	−0.26 *	0.004	−0.26 *	0.004	0.34 *	0.001	−0.35 *	0.001	−0.17 *	0.055

r_p_—Pearson correlation coefficient for parametric distributions; r_s_—Spearman* correlation coefficient for nonparametric distributions; *p*—statistical significance level; 25(OH)D—concentration of 25-hydroxyvitamin D; Bio25(OH)D—concentration of bioavailable 25(OH)D calculated based on directly measured free25(OH)D; BMI —Body Mass Index; HDL—concentration of high-density lipoprotein; TG—concentration of triglycerides; FBG—concentration of blood glucose.

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
