# Peer review of "Evaluation of Vitamin D Fractions in Obese Hypertensive Patients"

_ijerph, 2020, doi:10.3390/ijerph17051660_

Round 1

Reviewer 1 Report

The role of Vitamin D in obese hypertensive patients is a topic worth investigating. The article is well written but the presentation of the results could be improved.

It would benefit the paper to also visualize the results of table 2 in form of boxplots with confidence intervals to highlight which biomarker better discriminates between the two populations.

I would be also interested in looking at the scatterplot of Calculated free25(OH)D vs FBG in the whole population. (It is not necessary to add it to the paper)

In Figure 1 there is a spelling mistake “Patients meeteing inclusion”

Reviewer 2 Report

General comments

This report consists of a case control descriptive transversal study. It is well structured, written and clearly spelled out. This study is an extension of an earlier study published in 2017 in Arch Med Sci by the same group. Interestingly when perusing table III of tat report, total 25(OH)D concentrations are either as or more informative than the different sub-fractions. In this report, the authors show that obesity is the major component influencing the different fractions of circulating [25(OH)D3]. They further show that none of the fractions have a diagnostic value when looking at different variables included in the metabolic syndrome (hypertension, dysglycemia, dyslipidemia). This has been reported before in other studies, although not with as much details in terms of the the 25(OH)D fraction characterization, and hence is somewhat confirmatory.

Specific comments

Line 84: The acronym NOBHT is misleading: It could mean non-obese hypertensive. It should be changed to NOBNHT. This applies to the heading in table 1.

Lines 110 and on

More details should be given regarding the analytical methods, particularly for those on 25(OH)D3. This should include the principles and the proficiency data (CV, accuracy, LOD etc,).

Lines 125-130: The equations for free calculating & bioavailable 25(OH)D3 use different expressions for the concentration of albumin. The first is in g/L whereas the second is in mmol/L. The later is the proper one as all other constituents of the equations are expressed as molar units. Furthermore the authors express the serum albumin concentration in molar units (table 1).

Shouldn’t the terms “103”x[albumin] and “108”x[VDBP] and “105”[mol-1] be expressed as exponents?

Although mathematically & theoretically interesting, the calculation or direct measurement of the free [25(OH)D3[ concentration brings little new information. If we consider that total [25(OH)D3] is the sum of the VDBP-bound, albumin-bound & free fractions accounting respectively for approximately 85%, 14% & 1%, for all practical purposes, the bioavailable fraction [total 25(OH)D3 – VDBP-bound fraction] gives the same information (± 1%). It could be argued that cost/benefit is not favourable to this exercise when considering clinical situations. This is understood in their conclusion.

Reviewer 3 Report

In this paper PelczyĹ„ska et al, analyzed vitamin D levels in a large group of obese hypertensive patients and to study associations with glucose levels, and lipid levels. Authors identified Vitamin D fractions are decreased in obese hypertensive subjects, and are associated with anthropometric parameters, but not with glucose level or lipid profile, thus they concluded vitamin D levels cannot be considered as a predictive markers of metabolic disorders in this group of patients. 

The study was conducted on large group of patients, and the provide important information in vitamin D levels in hypertensive subjects.
